# NEURAL RATE CONTROL FOR LEARNED VIDEO COMPRESSION

**Yiwei Zhang[1], Guo Lu[1✉], Yunuo Chen[1], Shen Wang[1], Yibo Shi[2], Jing Wang[2], Li Song[1✉]**
[1]Institute of Image Communication and Network Engineering, Shanghai Jiao Tong University
[2]Huawei Technologies, Beijing, China
`{6000myiwei,luguo2014,cyril-chenyn,wangshen22206,song_li}@sjtu.edu.cn,`
`{shiyibo, wangjing215}@huawei.com`

## ABSTRACT

The learning-based video compression method has made significant progress in recent years, exhibiting promising compression performance compared with traditional video codecs. However, prior works have primarily focused on advanced compression architectures while neglecting the rate control technique. Rate control can precisely control the coding bitrate with optimal compression performance, which is a critical technique in practical deployment. To address this issue, we present a fully neural network-based rate control system for learned video compression methods. Our system accurately encodes videos at a given bitrate while enhancing the rate-distortion performance. Specifically, we first design a rate allocation model to assign optimal bitrates to each frame based on their varying spatial and temporal characteristics. Then, we propose a deep learning-based rate implementation network to perform the rate-parameter mapping, precisely predicting coding parameters for a given rate. Our proposed rate control system can be easily integrated into existing learning-based video compression methods. Extensive experiments show that our approach can achieve accurate rate control with only 2% average bitrate error. Better yet, our method achieves nearly 10% bitrate savings compared to various baseline methods.

## 1 INTRODUCTION

In recent years, video content has come to account for almost 80% of all internet traffic (Cisco, 2020). Therefore, it is crucial to design efficient video compression methods for video storage and transmission. Traditional video coding standards such as AVC (Wiegand et al., 2003), HEVC (Sullivan et al., 2012), and VVC (Ohm & Sullivan, 2018) have been manually designed over the past few decades based on block-partition, linear discrete cosine transform (DCT), and other methods.

Recently, there has been a growing interest in learning-based video compression methods. Existing methods (Lu et al., 2019; Agustsson et al., 2020; Hu et al., 2021; Lu et al., 2022; Sheng et al., 2022; Li et al., 2021; Shi et al., 2022; Li et al., 2022a; Xiang et al., 2023; Li et al., 2023) typically employ deep neural network to achieve motion compensation and residual/condition coding and optimize all the modules in the End-to-End compression framework.

Most existing learning-based video compression methods have not yet integrated rate control, a technique commonly used in practical applications. Traditional codecs use rate control to align the size of the encoded bitstream more closely with the target bitrate. This approach also boosts overall compression efficiency by allocating appropriate bitrates to various frames.

Unfortunately, for many of current learning-based video compression methods, the learned codecs are still primarily optimized under a single R-D trade-off point (fixed $\lambda$). While some approaches can implement variable bitrate coding in a single model (Choi et al., 2019; Yang et al., 2020a; Cui et al., 2021; Rippel et al., 2021; Li et al., 2022a), they require multiple rounds of compression to search for suitable coding parameters (usually the $\lambda$ parameter) to attain the desired bitrate. Additionally, even if we implement variable bitrate coding for rate control directly, existing learned video compression techniques fail to comprehensively address the issue of rate allocation during the rate control process, resulting in suboptimal compression efficiency.

One possible solution is to adopt traditional rate control methods, but these methods depend on empirical mathematical models to fit the relationship between bitrate and coding parameters, which may not be suitable for learning-based video compression methods. Moreover, traditional video codecs use pre-defined weights for rate allocation, without taking into account spatial and temporal content characteristics. Hence, it is necessary to develop a new rate control system for learned video compression methods.

Therefore, in this paper, we propose the first fully deep learning-based rate control system for learned video compression. Our proposed system is composed of two key components: a rate allocation network and a rate implementation network. Specifically, for a given bitrate budget of a video sequence, the rate allocation network will extract the corresponding spatiotemporal features to allocate the optimal bitrate for each frame according to its importance. Then the rate implementation network predicts proper coding parameters, such as the trade-off parameter $\lambda$ in our method, for each frame to achieve its target bitrate. Finally, we can precisely encode the video sequences at the given target bitrate. Meanwhile, thanks to the content adaptive rate allocation, we can further improve the overall video compression performance. Our proposed method is general and can be easily integrated with the existing video compression methods. To demonstrate the effectiveness of the proposed method, we apply our approach to four baseline methods Lu et al. (2019); Hu et al. (2021); Li et al. (2021); Shi et al. (2022) and perform extensive experiments on commonly used video benchmark datasets. Experimental results show that our approach can achieve accurate rate control with only 2% average bitrate error. Furthermore, the proposed method further brings nearly 10% bitrate saving compared to the baseline methods.

Our contributions are summarized below:

- We propose a general rate control approach for the learning-based video compression methods consisting of a rate allocation network and a rate implementation network. To the best of our knowledge, this is the first fully neural network-based rate control approach for learned video compression.

- Our plug-and-play rate control technique is simple but effective, achieving improved compression performance and accurate rate control on different learned video codecs.

## 2 RELATED WORKS

### 2.1 VIDEO COMPRESSION

Over the past decades, traditional video compression standards such as H.264(AVC) (Wiegand et al., 2003), H.265(HEVC) (Sullivan et al., 2012) and H.266(VVC) (Ohm & Sullivan, 2018) have been developed based on hybrid coding frameworks. The core modules, including inter-frame prediction, transformation, quantization, and entropy coding, have been well exploited to improve compression efficiency. By incorporating the rate control module, traditional coding standards can effectively ensure that the output bitrate closely matches the target bitrate, making them extensively applicable in diverse practical scenarios.

In recent years, deep learning-based video compression methods have evolved rapidly, showing promising results (Lu et al., 2019; Lin et al., 2020; Yang et al., 2020b; Hu et al., 2020; Yang et al., 2021; Hu et al., 2021; Li et al., 2021; Yang et al., 2022; Chang et al., 2022; Lin et al., 2022; Mentzer et al., 2022; Sheng et al., 2022; Li et al., 2022a; 2023). Lu et al. (2019) proposed a full learning-based video compression method DVC. It was based on a hybrid coding framework, in which all modules were replaced with deep learning to implement an end-to-end training process. To obtain a more accurate predicted frame, Lin et al. (2020) proposed using multi-frame information to predict the current reference frame. Agustsson et al. (2020) designed the scale-space flow to effectuate a more efficient alignment of the reference frame onto the current frame to be encoded. Yang et al. (2022) proposed an in-loop frame prediction method to predict the target frame in a recursive manner and achieve accurate prediction. Chang et al. (2022) proposed using the way of double-warp to derive the optical flow required for motion compensation by integrating the incremental and extrapolated optical flows. Besides, to enhance the residual coding performance, Hu et al. (2021) proposed to perform motion compensation and residual coding in the feature domain. Li et al. (2021) replaced the residual subtraction computation with a conditional coding strategy.

## 2.2 RATE CONTROL

Rate control is a highly beneficial tool in video coding, particularly in bandwidth-limited scenarios. In traditional video coding standards, rate control methods establish a mapping between the bitrate and encoding parameters and achieve the specified bitrate with minimal error.

There has been extensive research on rate control for traditional video coding standards, such as the R-Q (Ma et al., 2005; Liang et al., 2013), R-$\rho$ (Wang et al., 2013; Liu et al., 2010), and R-$\lambda$ (Li et al., 2014; 2016) models. Both the R-Q and R-$\rho$ models use the quantization parameter (QP) as the most critical factor determining the bitrate. The R-Q model establishes the relationship between the bitrate and the QP, using a quadratic function for fitting. The R-$\rho$ model establishes the relationship between the bitrate and the percentage of zero values in the quantization coefficient $\rho$ and models it as a linear function. However, with the development of various tools in traditional coding standards, QP is no longer the decisive factor in determining the bitrate.

To search for a more robust mathematical model for controlling the rate, Li et al. (2014) proposed to establish a mapping between the bitrate and the slope $\lambda$ of the rate-distortion (R-D) curve. Based on the fitting results of a large amount of data, the R-D relationship conforms to a hyperbolic model, and the relationship between R and $\lambda$ can be expressed as the derivative of the R-D relationship (Li et al., 2014). For various types of video content, the corresponding R-$\lambda$ model exhibits varying fitting parameters. Thus, in order to accommodate different content, the fitting parameters of the R-$\lambda$ model must be updated dynamically during the encoding process using a method similar to gradient descent. Thanks to its precise rate control effect, the R-$\lambda$ model is still utilized in traditional video coding standards. Additionally, some research has explored using learning-based methods in the rate control of traditional codecs. These methods (Hu et al., 2018; Mao et al., 2020) employ neural networks or reinforcement learning to predict the optimal quantization parameters in traditional codecs for each frame or coding unit. These methods are designed for traditional coding frameworks and may not be applicable to deep learning-based video coding schemes.

For learned video compression, Li et al. (2022b) proposed a rate control scheme for learned video compression similar to the traditional method. They attempted to establish an R-D-$\lambda$ analytical mathematical model, using the hyperbolic functions in Li et al. (2014) for approximation in order to achieve the mapping between rate and input variable rate parameter of the compression model. Besides, they also modeled the inter-frame dependency relationship as linear to derive the optimal rate allocation. Nevertheless, empirical mathematical models are derived from statistical analysis of large amounts of coding data of traditional codecs, and may not be applicable to learning-based video compression methods, thereby failing to achieve sufficiently accurate rate control. Xu et al. (2023) proposed a pixel-level rate allocation method that utilizes back-propagation through gradient ascent to find the optimal allocation strategy. However, this method needs multiple iterations and is unable to address the allocation approach in scenarios with a limited bitrate.

## 3 METHODOLOGY

### 3.1 SYSTEM FRAMEWORK

Let $\mathcal{X} = \{X_1, X_2, ..., X_t, X_{t+1}\}$ denote a video sequence, where $X_t$ represents a frame at time $t$. It is known that the existing learned video codecs are usually optimized by rate-distortion trade-off, *i.e.,* $R + \lambda D$. Here, $R, D$ represent the rate and distortion. $\lambda$ is the trade-off hyper-parameter. To enable continuous and precise rate control, the video codec should be capable of achieving variable bit rates through a single model. Therefore, we enhance the existing learned video codecs Lu et al. (2019); Hu et al. (2021); Li et al. (2021); Shi et al. (2022) with the off-the-shelf variable bitrate solution Lin et al. (2021) as our baseline methods in our proposed rate-control framework.

In rate control, considering the need to handle multiple levels of bitrates, we use the symbol $R$ with subscripts $s$, $mg$, and $t$ to denote the sequence level, mini group of pictures (miniGoP) level, and frame level bitrates, respectively. Bitrates with a superscript hat represent the actual encoded bitrates, while symbols without a superscript denote target bitrates. Fig. 1 shows the encoding process for the frame $X_t$ using our rate control strategy. We start by feeding consecutive video frames into the rate allocation network, assigning each frame the optimal rate allocation weight based on its spatial-temporal characteristics. Frames with larger weights are allocated with more bitrate and vice

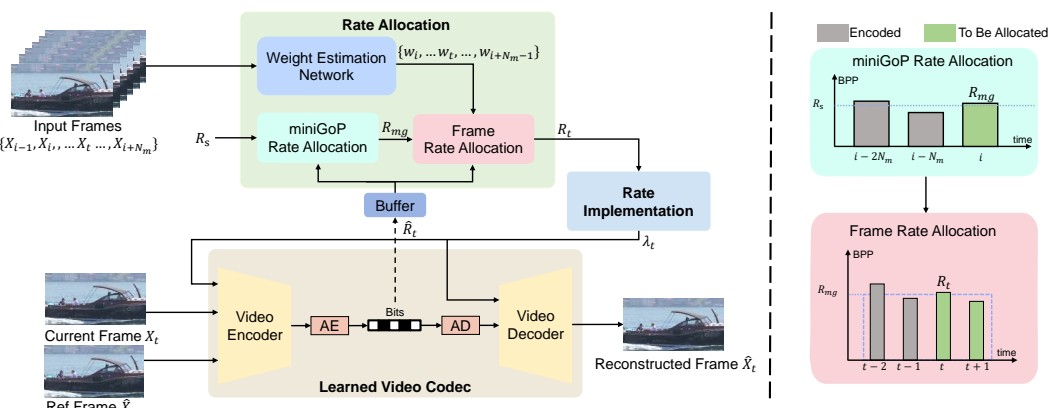

(a) Framework Overview                    (b) Rate Allocation Strategy

Figure 1: Figure (a) is an overview of our proposed neural rate control framework. Based on the given target bitrate $R_s$ and input frames, the rate allocation network produces target bitrate $R_t$ for the current frame $X_t$. Then the rate implementation module builds a mapping between bitrate $R_t$ and coding parameter $\lambda_t$, which is used for the learned video codec to encode $X_t$. Figure (b) is the visualization of our proposed two-level rate allocation strategy.

versa. According to the sequence-level target bitrate $R_s$ and remaining bitrate budget, we apply a two-level rate allocation to determine the target bitrate $R_t$ for $X_t$. Next, the rate implementation network maps $R_t$ to the predicted $\lambda_t$ for encoding $X_t$. The learned codec then compresses $X_t$ using $\lambda_t$, allowing precise rate control.

## 3.2 RATE ALLOCATION NETWORK

As shown in Fig. 1, our system allocates bitrates at two levels, namely the miniGoP level and the frame level. For the current frame $X_t$, the corresponding miniGoP includes a set of frames $\{X_i, X_{i+1}, ..., X_t, ..., X_{i+N_m-1}\}$. $N_m$ denotes the length of a miniGoP. During the miniGoP level rate allocation process, we first allocate bitrate to each miniGoP based on a uniform weight ratio in the following way,

$$R_{mg} = \frac{R_s \times (N_{coded} + SW) - \hat{R}_s}{SW} \times N_m \tag{1}$$

where $R_{mg}$ is the target bitrate for the current miniGoP, $R_s$ is the target average bitrate for the whole video sequence, $N_{coded}$ represents the number of frames that have been encoded, $\hat{R}_s$ is the total bitrate already consumed by the current encoding sequence. $SW$ refers to the sliding window size, which is used to ensure a smoother bitrate transition for each miniGoP during the encoding process. We set SW to 40 in our implementation.

As for the frame-level rate allocation within a miniGoP, we employ weights generated by the weight estimation network based on the spatiotemporal characteristics of the frames in this miniGoP. The allocation equation is shown in equation 2,

$$R_t = \frac{R_{mg} - \hat{R}_{mg}}{\sum_{j=t}^{i+N_m-1} w_j} \times w_t \tag{2}$$

where $R_t$ refers to the target bitrate required for frame $X_t$, $\hat{R}_{mg}$ represents the bitrate already consumed when encoding the current miniGoP, and $w_t$ denotes the rate allocation weight obtained from the weight estimation network for $X_t$. After that, we can get the target bitrate for the current frame $X_t$ to achieve optimal rate allocation given the overall target bitrate $R_s$. After the $t$-th frame is encoded, the actual encoded rate $\hat{R}_t$ will be updated in the buffer for the calculation of $\hat{R}_s$ and $\hat{R}_{mg}$.

Fig. 2 (a) shows the structure of our proposed weight estimation network. We use a lightweight network architecture by using several convolution and MLP networks. The convolution network extracts spatiotemporal features from a set of temporal consecutive frames, while the full connection network modulates the features extracted by the convolution network based on information obtained from the encoded results. Specifically, the input of the model consists of the frames in the current

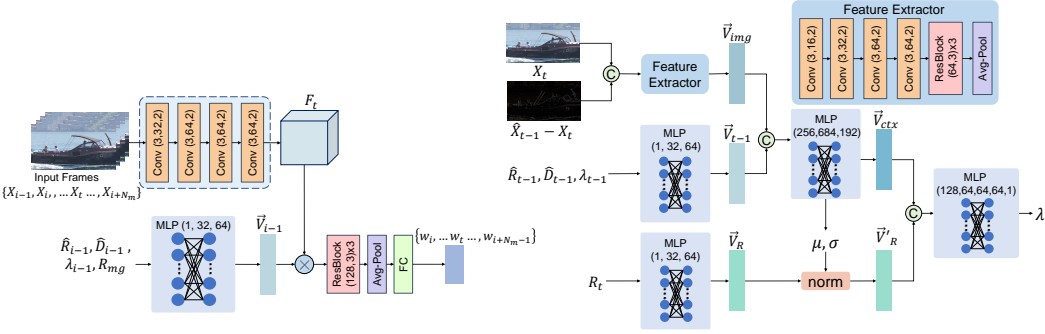

(a) Weight Estimation Network      (b) Rate Implementation Network

Figure 2: The network structure of the weight estimation network and rate implementation network.

miniGoP, as well as the frames $(X_{i-1}, X_{i+N_m})$ before and after the current miniGoP. We use the convolution network to extract the corresponding spatiotemporal features $F_t$. Besides, we further introduce the critical statistical information from the previous time step, including the bitrate $\hat{R}_{i-1}$, distortion $\hat{D}_{i-1}$ and the $\lambda_{i-1}$, along with the target bitrate for the current miniGoP. Here, we use the MLP networks to extract the corresponding feature vector, which is fused with features $F_t$ through channel-wise multiplication. Finally, the fused features are refined by the Resblocks and fully connected layers to generate the weights $[\omega_i, ..., \omega_t, ..., \omega_{i+N_m-1}]$ for each frame in a miniGoP.

The purpose of including the input $X_{i-1}$, its encoding results and $R_{mg}$ is to account for the influence of the previous reference frame on the current miniGoP. If $X_{i-1}$ is a relatively high-quality frame, then a lower bit rate will be used to encode the front part of the current miniGoP, and the overall quality will not decrease significantly due to the high-quality reference frame.

### 3.3 Rate Implementation Network

The rate implementation network aims to build a mapping between rate $R$ and coding parameter $\lambda$. Hence, one straightforward solution is to use MLP layers to model this relationship. However, considering the variable video content, this straightforward solution may not work well. In our implementation, we formulate the mapping as a regression problem conditioned on the content of the current frame to be coded and the encoding results of the previous frame.

Fig. 2 (b) shows the detailed architecture for our rate implementation network. In our proposed approach, we further introduce the content information from the current frame and statistical coding information from the previous frame to achieve content-adaptive R-$\lambda$ mapping. Specifically, the current frame $X_t$ and the difference map between $X_t$ and the previous reconstructed frame $\hat{X}_{t-1}$ are used as inputs to the convolution network. After several convolutions and the average pooling, the image feature vector $\vec{V}_{img}$ is obtained. Meanwhile, the statistical coding information from the previous frame including the actual bitrate $\hat{R}_{t-1}$, the distortion $\hat{D}_{t-1}$ and the estimated coding parameter $\lambda_{t-1}$ are fed into an MLP network to produce the feature vector $\vec{V}_{t-1}$.

Due to the varying content of videos, the different input bitrates for different content in the rate implementation network may lead to similar output $\lambda$. Therefore, we implement a normalization module to normalize the input bitrate for better prediction accuracy. We fuse vectors $\vec{V}_{img}$ and $\vec{V}_{t-1}$ to produce the normalization parameter $(\mu, \theta)$ to modulate the original feature $\vec{V}_R$ from input target bitrate $R_t$ in Equation 3, where $\vec{V}'_R$ represents the normalized feature and will be used to predict the coding parameter $\lambda_t$ for the current frame $X_t$.

$$\vec{V}'_R = \frac{\vec{V}_R - \mu}{\theta} \tag{3}$$

### 3.4 Training Strategy

**Step-by-Step Training.** Our method consists of multiple distinct modules, each with different training objectives and interdependent relationships. The training of the rate allocation network relies

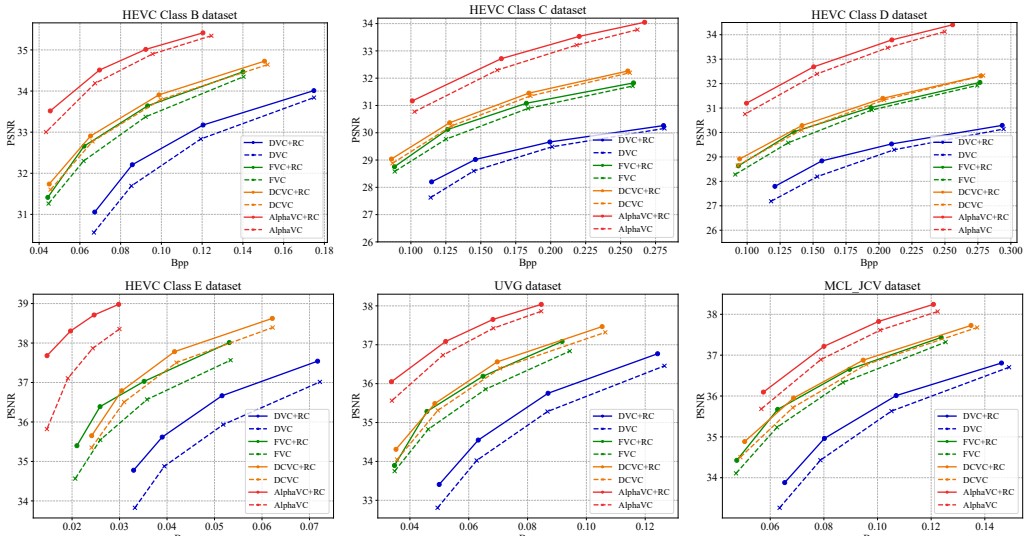

Figure 3: The R-D (rate-distortion) performance on HEVC Class B, Class C, Class D, Class E, UVG, and MCL_JCV datasets. Ours(DVC+RC), Ours(FVC+RC) and Ours(AlphaVC+RC) represent the methods integrated with our proposed rate control framework on DVC (Lu et al., 2019), FVC (Hu et al., 2021), DCVC (Li et al., 2021) and AlphaVC (Shi et al., 2022) baselines, respectively.

on an accurate rate implementation network. Therefore, we propose a step-by-step training strategy. First, we train a variable rate learned video compression method based on the modulated methods in Lin et al. (2021). The variable rate approach can be used for different baseline methods like DVC (Lu et al., 2019), FVC (Hu et al., 2021) DCVC (Li et al., 2021) and AlphaVC (Shi et al., 2022). We follow the default settings to train the variable rate learned codecs.

Then, we fix the parameters of the learned video codec and only train the rate implementation network to achieve a precise mapping model from the target rate to the encoding parameter $\lambda$. Specifically, the rate implementation network (*RI*) predicts coding parameters $\lambda_t$ for the $t$-th frame based on the target bit rate $R_t$. Our aim is to minimize the error between the target bitrate $R_t$ and actually encoded bitrate $\hat{R}_t$, which is obtained by the learned codec $C(\cdot)$ using the predicted coding parameters $\lambda_t$. Therefore, the loss function for training the rate implementation network is formulated in the following way,

$$L_{RI} = ((R_t - \hat{R}_t)/R_t)^2, where \ \hat{R}_t = C(\lambda_t) = C(RI(R_t)) \tag{4}$$

Finally, in the third step, we only train the rate allocation network while keeping the other parts of the model fixed. For the rate allocation network, it allocates weights for the frames in a miniGoP based on the frames within the miniGoP and its adjacent frames. During the training procedure, considering the error propagation effect when encoding multiple consecutive P frames, the loss function of the rate allocation network includes the rate-distortion loss of frames in $n$ miniGoPs and the neighboring frames. Therefore, the loss $L_{RA}$ for training the rate allocation network is formulated in the following way,

$$L_{RA} = \sum_{i=t}^{t+n*N_m} R_i + \lambda_g D_i \tag{5}$$

Where $R_i$ and $D_i$ represent the rate and distortion for frame $X_i$. $n$ denotes the number of miniGoPs, and $\lambda_g$ denotes the global lambda for training the current miniGoP. During the training stage, we randomly select a value for $\lambda_g$ and pre-encode one frame of the miniGoP using this value. The corresponding bitrate is then set as the target bitrate for training the rate allocation network.

Table 1: The relative bitrate error $\Delta R$ (%) and the BD-rate gain results (%) on the testing datasets.

| Dataset | $\Delta R$ (%) $\downarrow$ | | | | BD-rate (%) $\downarrow$ | | | |
|---|---|---|---|---|---|---|---|---|
| | DVC | FVC | DCVC | AlphaVC | DVC | FVC | DCVC | AlphaVC |
| HEVC B | 1.35 | 1.88 | 2.32 | 3.68 | -10.99 | -9.59 | -5.88 | -10.76 |
| HEVC C | 1.18 | 1.06 | 1.94 | 1.54 | -10.63 | -8.26 | -4.42 | -11.00 |
| HEVC D | 1.91 | 2.44 | 2.11 | 1.67 | -12.17 | -6.90 | -3.80 | -8.94 |
| HEVC E | 1.11 | 1.86 | 1.33 | 1.19 | -18.28 | -20.03 | -9.24 | -33.90 |
| UVG | 2.82 | 2.86 | 2.80 | 0.61 | -11.61 | -12.33 | -7.34 | -12.28 |
| MCL_JCV | 2.79 | 2.62 | 2.95 | 1.17 | -8.78 | -9.37 | -5.68 | -8.69 |
| Average | 1.86 | 2.12 | 2.24 | 1.64 | -12.08 | -11.08 | -6.06 | -14.26 |

## 4 EXPERIMENTS

### 4.1 EXPERIMENTAL SETUP

**Training Datasets.** For training the rate implementation network, we used the Vimeo-90k dataset (Xue et al., 2019), containing 89,800 video clips. For the rate allocation network, we selected the BVI-DVC dataset (Ma et al., 2021) to leverage the rate-distortion loss of multiple frames. This dataset includes 800 video sequences of various resolutions, each with 64 frames. We trained the network using randomly cropped 256×256 patches from these video sequences.

**Evaluation Datasets.** We tested the performance of our algorithm on the HEVC standard test sequences (Class B, C, D, E) (Wiegand et al., 2003). This HEVC dataset contains 16 videos with diverse content characteristics and resolutions. Following the evaluation settings in the existing learned codecs, we also included the UVG (Mercat et al., 2020) and MCL_JCV (Wang et al., 2016) datasets in our experiments. For all baseline models and our proposed rate control methods, we set the GOP size to 100 during the evaluation stage.

**Evaluation Metrics** To evaluate compression performance on the benchmark datasets, we used Peak Signal-to-Noise Ratio (PSNR) against bits per pixel (bpp) as metrics. We also employed the BD-rate metric (Bjontegaard, 2001) for overall compression performance comparison. For assessing the accuracy of rate control, we utilized the relative bitrate error. This error, $\Delta R$, is defined as $\Delta R = |R_s - \hat{R}s|/Rs$, representing the discrepancy between the actual bitrate $\hat{R}s$ produced by the codec and the target bitrate $Rs$.

**Implementation Details** We reimplemented the DVC (Lu et al., 2019), FVC (Hu et al., 2021), DCVC (Li et al., 2021) and AlphaVC (Shi et al., 2022) as our baseline models. Since our method primarily focuses on rate control for P frames, we have excluded the condition I frame from AlphaVC (Shi et al., 2022). We employed the method in Lin et al. (2021) to enable continuous variable rate for these baseline methods. Other state-of-the-art learned video compression methods can also be integrated with our proposed rate control approach. In terms of our rate implementation network, we randomly selected variable rate parameters for encoding and input the resulting bitrate as the target bitrate for training. As for the rate allocation network, we set $n = 2$ and updated parameters by computing the rate-distortion loss of two consecutive miniGoPs along with their previous and subsequent frames. Both networks were trained over 200,000 steps, with a batch size of 4. The learning rate starts at 1e-4, reducing to 1e-5 after 120,000 steps. The training times for the rate implementation and allocation networks are about 10 hours and 1 day, respectively. During inference for the first P frame, we use a default rate and distortion value ($R = 1, D = 0$) to indicate the preceding I-frame had a high rate and low distortion. For subsequent P frames, we use the rate and distortion of the previously coded frame.

### 4.2 EXPERIMENTAL RESULTS

**Performance** Fig. 3 provides the rate-distortion performance over the evaluation datasets for different compression methods. For the baseline models, we assessed compression performance at four $\lambda$ points, namely $\lambda s = \{256, 512, 1024, 2048\}$. And the corresponding actual bitrate in each sequence was set as the target bitrate for our proposed rate control based video compression system. Therefore, we had a fair comparison with the baseline method at the same bitrate level.

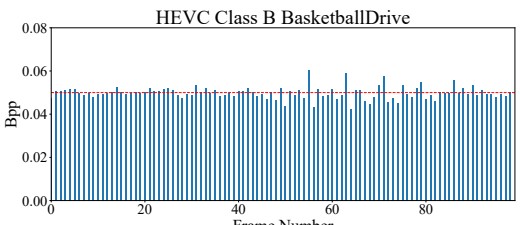
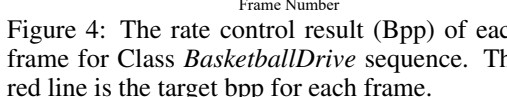
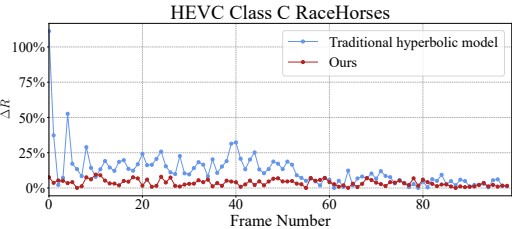

Figure 4: The rate control result (Bpp) of each frame for Class *BasketballDrive* sequence. The red line is the target bpp for each frame.

Figure 5: Comparison of rate error $\Delta R$ between our rate implementation network and the traditional hyperbolic model. The target bpp for each frame is set as 0.25 bpp.

It can be observed that our rate control framework achieves a bitrate that is relatively close to that of the baseline method using fixed $\lambda$ encoding. This indicates that our method can enable precise rate control. Quantitative results are shown in Table 1. The proposed method achieves an average 1% $\sim$ 3% rate error when compared with the target bitrate in different baseline methods and datasets.

Furthermore, our method can also bring an overall improvement in compression performance. The BD-rate savings are also presented in Table 1 and it is noted that our method achieves nearly 10% bitrate savings on average when compared with baseline methods. In particular, for Class E sequences with predominantly static scenes, our method attains more significant performance gains by adjusting the bitrate allocation, leading to 9% to 33% bitrate savings. The reason is that our rate control method allocates more bitrates to the important frames, which has a huge influence on the quality of subsequent reconstructed frames. In contrast, most existing frameworks use the same weights for each frame and may suffer from the cumulative error problem.

## 4.3 ABLATION STUDIES

**Rate Implementation Accuracy** To further show the accuracy of the proposed rate implementation network, we provide the bpp for each frame of HEVC Class B *BaseketballDrive* Sequence in Fig. 4. Here, we do not use the rate allocation network and allocate each frame in sequence with the same target bitrate. The results indicate that our method is able to encode each frame with very low bit rate errors. In detail, we set 0.05 bpp as the target bpp for each frame in the sequence. The corresponding actual average coding bitrate is 0.0499 and the average relative bitrate error is 0.21%.

**Effectiveness of Rate Allocation** The rate allocation network considers the spatiotemporal characteristics of different frames for optimal rate allocation and improves compression performance at a given bitrate. To validate our rate allocation approach, we conducted an experiment using fixed bitrates for each frame. e. As shown in Fig. 6, removing the rate allocation network (*Ours w/o RA*) significantly reduced the overall compression performance, indicating that uniform bitrate allocation across frames is suboptimal.

To further observe the role of rate allocation networks, Fig. 7 displays the variation in PSNR and bpp of different frames during the encoding process. The network mitigates quality degradation by dynamically adjusting bitrates for sequences of P frames, thus improving frame quality and minimizing cumulative errors. It can be observed that the rate allocation network adaptively assigns two high-quality frames in a miniGoP at the initial stage, while only one is given in the later stage.

**Analysis for Traditional Rate Control** In order to compare our method with the traditional rate control method based on empirical mathematical models presented in Li et al. (2022b), we utilized the same variable rate model on DVC (Lu et al., 2019) and reimplemented their method. We conducted experiments on HEVC Class C and D datasets under the GOP size of 100. Li's method (Li et al., 2022b) resulted in bitrate errors of 7.72% and 8.43% for Class C and D respectively, with performance decreases of 3.92% and 1.01%. In contrast, our method achieved significantly lower bitrate errors of only 1.18% and 1.91%, with performance gains of 10.63% and 12.17% respectively. Fig. 5 illustrates the frame-by-frame bitrate errors of our method and the hyperbolic R-$\lambda$ model on one HEVC Class C sequence. Our proposed rate implementation network achieves significantly smaller rate errors. Since the traditional method requires dynamic parameter updates of the hyperbolic model during the encoding process to achieve effective prediction, it exhibits substantial rate

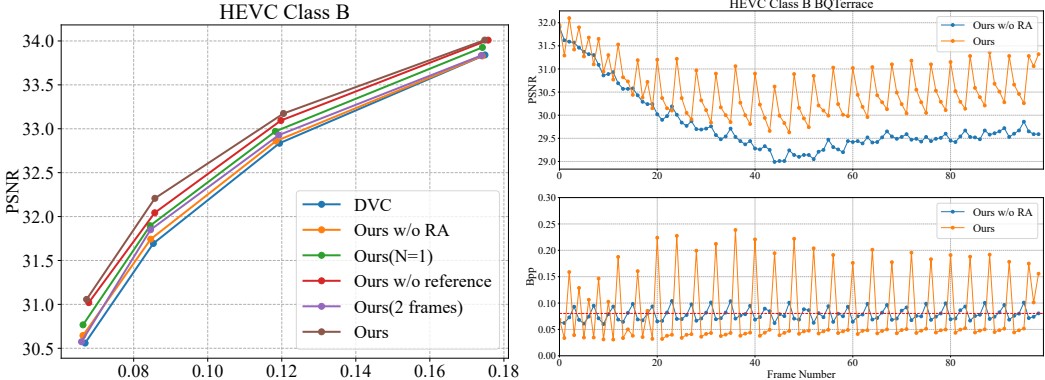

Figure 6: Ablation study on rate allocation network. The R-D performance is calculated on the HEVC Class B dataset.

Figure 7: The variation of PSNR and bpp during the encoding process. Ours w/o RA denotes the encoding results obtained without the rate allocation network.

errors at the initial encoding stage. In contrast, our method can achieve accurate prediction including the initial stages of encoding.

**Effectiveness of Different Components** Fig. 6 displays the further analysis of our rate allocation network. We first assessed the training loss, as defined in Equation 5. This loss function includes R-D (rate-distortion) losses for frames within two miniGoPs. For comparison, we also conducted experiments using fewer frames, specifically one miniGoP for training losses (denoted as *Ours(N=1)*). The results show that using R-D losses from more frames leads to notably enhanced performance improvements.

We also analyze the inputs for the rate allocation network. Experimental results show that omitting coding data (distortion, bitrate, *etc*) from the previous reference frame and the target bitrate for the current miniGoP (denoted as *Ours w/o reference*) leads to a 3.09% decrease in RD performance. Besides, reducing a miniGoP to 2 frames also lowers RD performance (*Ours 2 frames*). Conversely, increasing a miniGoP to 8 frames doubles both the parameters in the weight estimation network and training time, but only slightly improves RD performance by 0.12%. Hence, setting the miniGoP size to 4 represents a more optimal balance.

For the rate implementation network, we demonstrate the effectiveness of the normalization operation and the input frame information. Without normalization, using fully connected networks to predict coding parameters increases average rate errors on DVC (Lu et al., 2019) for HEVC Classes B, C, D, and E to 1.87%, 1.51 %, 2.69%, and 3.09%, respectively. As for the frame information, eliminating coding data from the reference frame causes training instability and hampers effective rate control. Removing the residual image increases average rate errors for HEVC Class B, C, D, and E datasets to 3.56%, 2.43%, 2.85%, and 3.96%.

**Running Time and Model Complexity** Our rate control framework adds operations only to the encoder, keeping the decoder's complexity unchanged from the original model. The rate allocation and implementation networks have 443K and 564K learnable parameters, respectively. When encoding a 1080P sequence, the inference times for these networks are just 2.95ms and 2.32ms, respectively.

## 5 CONCLUSIONS AND FUTURE WORKS

In this paper, we present the first fully deep learning-based rate control scheme for learned video codec. Our method consists of a rate implementation network and a rate allocation network to achieve precise rate control on several benchmark datasets using various baseline methods. Furthermore, thanks to the optimal bitrate allocation, we can further improve the overall compression performance at the target bitrate level. Our method is agnostic to the existing learning-based video compression method and only requires a small additional computational overhead on the encoding side. In the future, we will extend our rate control framework for bidirectional B-frame video compression or multiple reference frames video compression.

## ACKNOWLEDGEMENTS

This work was supported in part by National Natural Science Foundation of China(62102024,62331014), Fundamental Research Funds for the Central Universities, STCSM under Grant 22DZ2229005, 111 project BP0719010.

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

## A  MORE EXPERIMENTAL RESULTS ON RATE CONTROL

In Table 2, 3, 4 and 5, we provide the per sequence rate-distortion (R-D) performance and relative bitrate error on HEVC Class B, Class C, Class D, and Class E datasets.

## B  ADDITIONAL RESULTS FOR ABLATION STUDY AND ANALYSIS

In this section, we provide more results for the analysis of rate control accuracy and the effectiveness of the rate allocation network.

**Rate Control Accuracy** Fig. 4 shows the bpp for each frame in HEVC Class B *Cactus* sequence and HEVC Class D *BasketballPass* sequence. The target bitrate for *cactus* sequence is 0.15 bpp. The actual average coding bitrate is 0.1486 bpp and the average relative bitrate error is 0.93%. In addition, The target bitrate for *BasketballPass* sequence is 0.12 bpp. The average coding bitrate is 0.1205 and the average relative bitrate error is 0.45%.

**Effectiveness of Rate Allocation** Fig. 9 and 10 show the variation in PSNR and bpp of different frames during the encoding process. The red line represents the target bitrate. We also provide some comparison of the subjective quality of the reconstructed frame in Fig. 11.

Table 2: R-D performance and rate control accuracy on HEVC Class B dataset

| Sequence | DVC | | Ours(DVC+RC) | | | FVC | | Ours(FVC+RC) | | |
|---|---|---|---|---|---|---|---|---|---|---|
| | Bpp | PSNR | Bpp | PSNR | $\Delta R\%$ | Bpp | PSNR | Bpp | PSNR | $\Delta R\%$ |
| BasketballDrive | 0.07 | 31.47 | 0.08 | 32.13 | 3.41 | 0.05 | 32.14 | 0.05 | 32.37 | 0.51 |
| | 0.09 | 32.55 | 0.09 | 33.07 | 2.92 | 0.06 | 33.18 | 0.07 | 33.58 | 2.37 |
| | 0.12 | 33.63 | 0.13 | 33.93 | 2.05 | 0.09 | 34.12 | 0.09 | 34.38 | 3.19 |
| | 0.17 | 34.53 | 0.17 | 34.62 | 0.07 | 0.12 | 34.89 | 0.12 | 34.98 | 0.24 |
| BQTerrace | 0.07 | 28.91 | 0.07 | 29.60 | 2.08 | 0.05 | 29.64 | 0.05 | 29.81 | 0.43 |
| | 0.09 | 29.91 | 0.09 | 30.61 | 2.97 | 0.07 | 30.66 | 0.08 | 31.01 | 3.40 |
| | 0.13 | 30.97 | 0.14 | 31.37 | 1.92 | 0.12 | 31.76 | 0.12 | 32.06 | 4.70 |
| | 0.21 | 31.98 | 0.21 | 32.18 | 1.56 | 0.19 | 32.75 | 0.19 | 32.85 | 2.72 |
| Cactus | 0.06 | 30.09 | 0.06 | 30.55 | 0.96 | 0.04 | 30.52 | 0.04 | 30.61 | 2.96 |
| | 0.08 | 31.04 | 0.08 | 31.50 | 1.38 | 0.05 | 31.37 | 0.05 | 31.75 | 2.51 |
| | 0.11 | 31.98 | 0.11 | 32.29 | 0.05 | 0.08 | 32.31 | 0.08 | 32.56 | 2.18 |
| | 0.16 | 32.81 | 0.16 | 33.01 | 0.38 | 0.13 | 33.24 | 0.13 | 33.40 | 2.60 |
| Kimono1 | 0.06 | 32.74 | 0.06 | 33.11 | 0.11 | 0.04 | 33.80 | 0.04 | 33.91 | 0.81 |
| | 0.08 | 34.22 | 0.08 | 34.49 | 0.99 | 0.06 | 35.03 | 0.06 | 35.30 | 0.24 |
| | 0.10 | 35.63 | 0.11 | 35.83 | 0.54 | 0.08 | 36.22 | 0.08 | 36.33 | 0.97 |
| | 0.14 | 36.78 | 0.14 | 36.78 | 1.70 | 0.11 | 37.17 | 0.11 | 37.20 | 2.26 |
| ParkScene | 0.07 | 29.59 | 0.07 | 29.90 | 1.89 | 0.04 | 30.21 | 0.04 | 30.36 | 2.71 |
| | 0.09 | 30.75 | 0.09 | 31.38 | 1.36 | 0.06 | 31.26 | 0.06 | 31.66 | 2.09 |
| | 0.13 | 31.97 | 0.13 | 32.47 | 0.44 | 0.10 | 32.46 | 0.10 | 32.87 | 0.17 |
| | 0.19 | 33.11 | 0.19 | 33.46 | 0.20 | 0.15 | 33.69 | 0.15 | 33.91 | 0.46 |

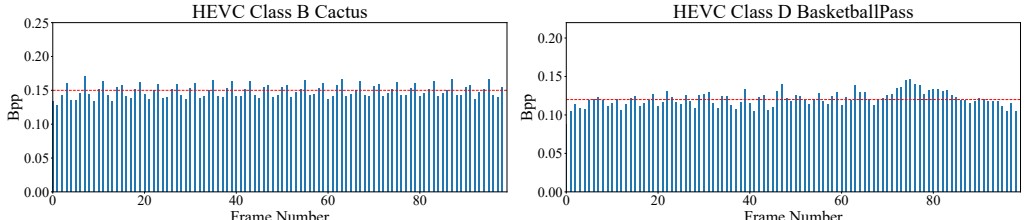

Figure 8: The rate control result (Bpp) of each frame for Class B *Cactus* and Class D *BasketballPass* sequences, respectively. The red line is the target bpp for every frame.

Table 3: R-D performance and rate control accuracy on HEVC Class C dataset

| Sequence | DVC | | Ours(DVC+RC) | | | FVC | | Ours(FVC+RC) | | |
|---|---|---|---|---|---|---|---|---|---|---|
| | Bpp | PSNR | Bpp | PSNR | $\Delta R\%$ | Bpp | PSNR | Bpp | PSNR | $\Delta R\%$ |
| BasketballDrill | 0.08 | 29.31 | 0.08 | 29.94 | 0.32 | 0.06 | 30.31 | 0.06 | 30.43 | 1.06 |
| | 0.10 | 30.39 | 0.10 | 30.89 | 0.92 | 0.08 | 31.48 | 0.08 | 31.88 | 0.38 |
| | 0.13 | 31.44 | 0.13 | 31.68 | 1.32 | 0.11 | 32.54 | 0.11 | 32.80 | 0.87 |
| | 0.19 | 32.33 | 0.18 | 32.44 | 0.85 | 0.15 | 33.37 | 0.15 | 33.58 | 0.39 |
| BQMall | 0.10 | 28.31 | 0.10 | 28.94 | 2.67 | 0.07 | 29.48 | 0.07 | 29.68 | 0.45 |
| | 0.12 | 29.32 | 0.12 | 29.81 | 1.30 | 0.10 | 30.69 | 0.10 | 31.13 | 1.67 |
| | 0.16 | 30.24 | 0.16 | 30.46 | 0.55 | 0.14 | 31.82 | 0.14 | 32.10 | 1.02 |
| | 0.23 | 30.88 | 0.23 | 31.11 | 0.85 | 0.20 | 32.71 | 0.20 | 32.87 | 0.11 |
| PartyScene | 0.14 | 25.25 | 0.14 | 25.75 | 1.66 | 0.12 | 26.12 | 0.12 | 26.34 | 1.78 |
| | 0.18 | 25.96 | 0.18 | 26.39 | 0.31 | 0.17 | 27.26 | 0.17 | 27.70 | 1.28 |
| | 0.26 | 26.51 | 0.25 | 26.76 | 1.60 | 0.25 | 28.43 | 0.25 | 28.69 | 1.07 |
| | 0.35 | 26.91 | 0.35 | 27.06 | 0.57 | 0.34 | 29.23 | 0.34 | 29.36 | 0.02 |
| RaceHorses | 0.14 | 27.64 | 0.14 | 28.18 | 2.05 | 0.10 | 28.41 | 0.11 | 28.57 | 1.96 |
| | 0.18 | 28.74 | 0.19 | 28.99 | 2.90 | 0.15 | 29.64 | 0.16 | 29.78 | 3.76 |
| | 0.26 | 29.76 | 0.25 | 29.75 | 0.36 | 0.24 | 30.78 | 0.24 | 30.74 | 0.27 |
| | 0.36 | 30.52 | 0.36 | 30.43 | 0.67 | 0.34 | 31.53 | 0.35 | 31.49 | 0.85 |

Table 4: R-D performance and rate control accuracy on HEVC Class D dataset

| Sequence | DVC | | Ours(DVC+RC) | | | FVC | | Ours(FVC+RC) | | |
|---|---|---|---|---|---|---|---|---|---|---|
| | Bpp | PSNR | Bpp | PSNR | $\Delta R\%$ | Bpp | PSNR | Bpp | PSNR | $\Delta R\%$ |
| BasketballPass | 0.09 | 29.30 | 0.10 | 30.06 | 9.74 | 0.07 | 30.49 | 0.08 | 30.99 | 6.56 |
| | 0.12 | 30.35 | 0.12 | 30.97 | 7.07 | 0.10 | 31.81 | 0.10 | 32.38 | 5.15 |
| | 0.16 | 31.53 | 0.15 | 31.70 | 1.78 | 0.14 | 33.07 | 0.14 | 33.23 | 1.20 |
| | 0.21 | 32.46 | 0.21 | 32.58 | 0.07 | 0.19 | 34.11 | 0.19 | 34.30 | 1.46 |
| BlowingBubbles | 0.11 | 27.18 | 0.11 | 27.73 | 1.18 | 0.08 | 27.87 | 0.08 | 28.20 | 1.44 |
| | 0.14 | 28.23 | 0.14 | 28.86 | 0.98 | 0.11 | 29.07 | 0.12 | 29.63 | 4.37 |
| | 0.19 | 29.31 | 0.19 | 29.58 | 2.05 | 0.17 | 30.38 | 0.16 | 30.61 | 2.53 |
| | 0.27 | 30.21 | 0.27 | 30.45 | 0.99 | 0.24 | 31.43 | 0.24 | 31.58 | 0.12 |
| BQSquare | 0.11 | 25.03 | 0.12 | 25.73 | 0.51 | 0.08 | 26.26 | 0.09 | 26.63 | 2.46 |
| | 0.15 | 25.78 | 0.15 | 26.66 | 1.12 | 0.13 | 27.54 | 0.13 | 27.88 | 4.44 |
| | 0.20 | 26.56 | 0.20 | 26.99 | 0.15 | 0.20 | 28.80 | 0.20 | 28.91 | 2.20 |
| | 0.29 | 27.08 | 0.29 | 27.32 | 0.67 | 0.28 | 29.66 | 0.28 | 29.81 | 0.85 |
| RaceHorses | 0.16 | 27.23 | 0.16 | 27.66 | 1.34 | 0.13 | 28.51 | 0.14 | 28.71 | 2.41 |
| | 0.21 | 28.43 | 0.22 | 28.86 | 2.03 | 0.19 | 29.90 | 0.19 | 30.22 | 2.17 |
| | 0.30 | 29.78 | 0.29 | 29.86 | 0.81 | 0.28 | 31.44 | 0.27 | 31.34 | 0.61 |
| | 0.41 | 30.81 | 0.41 | 30.82 | 0.03 | 0.39 | 32.50 | 0.39 | 32.51 | 1.07 |

Table 5: R-D performance and rate control accuracy on HEVC Class E dataset

| Sequence | DVC | | Ours(DVC+RC) | | | FVC | | Ours(FVC+RC) | | |
|---|---|---|---|---|---|---|---|---|---|---|
| | Bpp | PSNR | Bpp | PSNR | $\Delta R\%$ | Bpp | PSNR | Bpp | PSNR | $\Delta R\%$ |
| KristenAndSara | 0.03 | 33.75 | 0.03 | 34.84 | 0.68 | 0.02 | 34.60 | 0.02 | 35.48 | 1.78 |
| | 0.04 | 34.77 | 0.04 | 35.58 | 1.56 | 0.03 | 35.56 | 0.03 | 36.59 | 1.66 |
| | 0.05 | 35.84 | 0.05 | 36.62 | 1.11 | 0.04 | 36.64 | 0.04 | 37.17 | 1.72 |
| | 0.07 | 36.91 | 0.07 | 37.44 | 0.71 | 0.05 | 37.62 | 0.05 | 38.13 | 0.21 |
| FourPeople | 0.04 | 33.44 | 0.04 | 34.43 | 1.67 | 0.02 | 34.01 | 0.02 | 34.51 | 2.25 |
| | 0.04 | 34.60 | 0.04 | 35.17 | 2.96 | 0.03 | 35.07 | 0.03 | 35.45 | 3.95 |
| | 0.06 | 35.71 | 0.06 | 36.22 | 1.25 | 0.04 | 36.07 | 0.04 | 36.48 | 3.54 |
| | 0.08 | 36.83 | 0.08 | 37.35 | 1.22 | 0.06 | 37.10 | 0.06 | 37.52 | 1.98 |
| Johnny | 0.03 | 34.29 | 0.03 | 35.07 | 0.30 | 0.02 | 35.09 | 0.02 | 36.20 | 2.16 |
| | 0.04 | 35.27 | 0.04 | 36.10 | 0.73 | 0.03 | 36.01 | 0.03 | 37.12 | 1.57 |
| | 0.05 | 36.25 | 0.05 | 37.15 | 0.69 | 0.03 | 37.00 | 0.03 | 37.43 | 1.34 |
| | 0.07 | 37.31 | 0.07 | 37.83 | 0.38 | 0.05 | 37.97 | 0.05 | 38.38 | 0.10 |

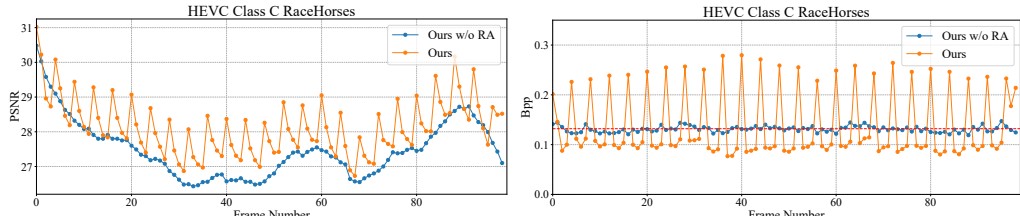

Figure 9: The variation of PSNR and bpp during the encoding process for HEVC Class C *Race-Horses* sequence. w/o RA denotes the encoding results obtained without the rate allocation network.

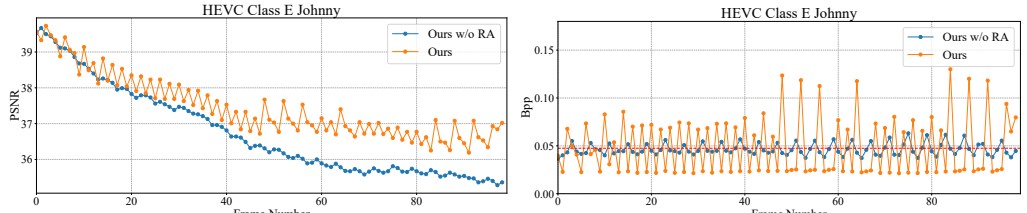

Figure 10: The variation of PSNR and bpp during the encoding process for HEVC class E *Johnny* sequence. Ours w/o RA denotes the encoding results obtained without the rate allocation network.

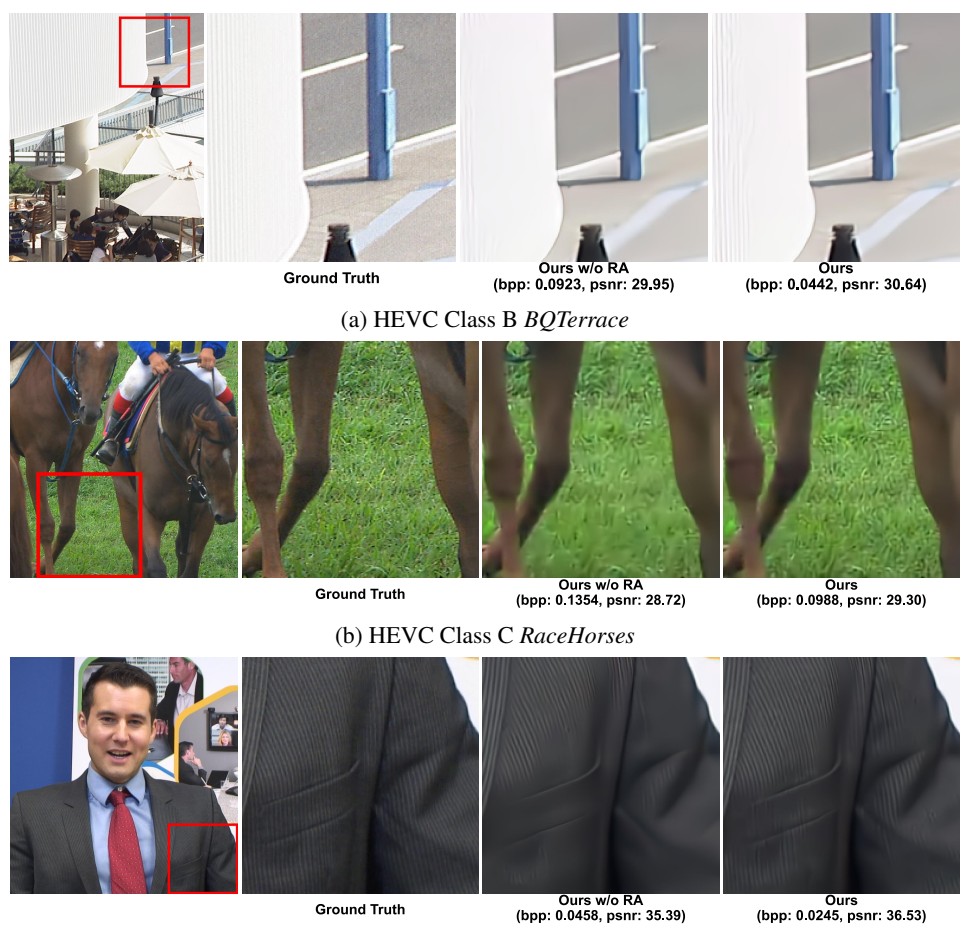

(a) HEVC Class B *BQTerrace*

(b) HEVC Class C *RaceHorses*

(c) HEVC Class E *Johnny*

Figure 11: Visual quality comparison between our approach with and without rate allocation (RA) network.

