# OpenReview forum: "Neural Rate Control for Learned Video Compression"
_ICLR.cc/2024/Conference — ICLR 2024 poster_

### Official Review · Reviewer_gAxb · 2023-10-19

**Soundness:** 3 good
**Presentation:** 4 excellent
**Contribution:** 4 excellent
**Rating:** 8
**Confidence:** 5

**Summary:**

This is a paper that describes a method to add adaptive rate control to a variable rate neural video codec.

IIUC it works as follows:

1. train a NVC with variable lambda support.
2. train a "rate implementation network" that can predict a lambda matching some target rate R_t.
3. train a "rate allocation network" that predicts R_t such that we get good rate distortion characteristics over a group of frames (Eq 5).

**Strengths:**

The authors present their idea well, and it was relatively easy to understand (although it would have been nice to have a high level summary of how the components are trained before going into the details in Sec. 3.4, eg., a list like what I wrote in "Summary" above).

The method is ablated on multiple baseline methods, and achieves significant gains throughout.

Various parts of the method are ablated and shown to be effective.

Overall, the paper has a clear simple idea that is easy to follow, and shows that it works well.

**Weaknesses:**

My only gripe is it is a bit hard to follow the details and notation, since a lot of symbols are introduced (for example, we have R_mg, R_tar, R_coded, R_t, \hat R_t, R_coded_m). Not all are wlel introduced (eg \hat R_t was only used in the figure before it appeared in the text).

I think the clarity of the text could be improved by either simplyfying the notation, or replacing some of the notation with a description.

**Questions:**

It was unclear to me why we need two stages to trainallocation and implementation. Could we not train them jointly? Basically one blackbox that takes as input the R_tar (target over group of frames) and predicts \lambda t such that \hat R_t is as desired.

---

> ### Author Response · Authors · 2023-11-17
>
> Thank you for your insightful questions. We have prepared detailed, point-by-point responses to each query. We hope this addresses your concerns effectively.
>
> **Q1: About the notation complexity**.
>
> **A1:** Thanks for your suggestion, and very sorry for the inconvenience we brought to you. To solve this issue, we first standardized the usage of symbols. Since our method allocates bitrate in three levels: sequence level, miniGoP level, and frame level, we need different symbols for the target rate and consumed rate, which results in complicated notations.
>
> To make it clear, we will provide one symbol table for different rate symbols at the beginning of the Method part. The origin symbol table and revised table are as follows. In the tables, from top to bottom, there are rate symbols in sequence level, miniGoP level, and frame level. For the revised symbols, the subscript "s" represents the bitrate at the sequence level, the subscript "m" represents the bitrate at the minigop level, and the subscript "t" represents the bitrate at the frame level. The symbol with a bar indicates the accumulated bitrate consumed at the sequence and minigop levels. The symbol with a hat superscript, $\hat{R_t}$, represents the actual encoded bitrate of the current frame.
>
>
> | Component                        | Original Symbol | Revised Symbol |
> |----------------------------------|-----------------|----------------|
> | Target bitrate for sequence      | $R_{tar}$       | $R_s$          |
> | Consumed bitrate for sequence    | $R_{coded}$     | $\bar{R}_s$    |
> | Target bitrate for miniGoP       | $R_{mg}$        | $R_m$          |
> | Consumed bitrate for miniGoP     | $R_{coded_m}$   | $\bar{R}_m$    |
> | Allocated bitrate for t-th frame | $R_t$           | $R_t$          |
> | Actual bitrate for t-th frame    | $\hat{R_t}$     | $\hat{R_t}$    |
>
>
> **Q2: About the "two-stage" model and training of our method.**
>
> **A2:** Thanks for your question. The optimization of the two modules involves different numbers of frames. The rate implement module aims to achieve the most accurate mapping relationship for a single frame. In contrast, rate allocation involves multi-frame training, where the loss of multiple frames is averaged and propagated together to obtain the optimal weight allocation results for multiple frames.
>
> As a result, the optimization of multi-frame rate-distortion loss may not be optimal for the rate implementation network. We attempted to train the two modules together as one black box, but the model could not be successfully trained.
>
> To address this issue, we have implemented a progressive training strategy. Initially, we trained the rate implementation network. Subsequently, in the next stage, we trained the rate allocation network while keeping the parameters of the rate implementation network fixed.

---

### Official Review · Reviewer_cdDb · 2023-10-31

**Soundness:** 3 good
**Presentation:** 2 fair
**Contribution:** 2 fair
**Rating:** 6
**Confidence:** 4

**Summary:**

This paper proposes a rate control method for learning based video compression. The proposed method is plug-and-play and consists of a rate allocation network and a rate implementation network. Experiments on multiple baseline models show that this method can accurately control the output bitrate. In addition, benefiting from more reasonable rate allocation, this method can also bring certain performance improvements.

**Strengths:**

1. The most important contribution of this paper is to propose a framework for designing rate control models for learning based video compression. And it is proved that this framework design is better than the rate control strategy designed based on empirical mathematical models.
2.	This paper demonstrates the broad applicability of the framework and provides a reasonable training method.
3.	The paper is clearly and understandably presented.

**Weaknesses:**

1.	The ablation of specific module design is not very sufficient. Could you give an ablation to explain the impact of introducing frame information?
2.   It's better to show the performance impact of different miniGoPs in the experimental section.

**Questions:**

1. Why there is a quality fluctuation with a period of 4 in figure 7? Is this related to the hyperparameter settings of miniGoP?
2. In figure 7, compared to the method without rate allocation, the code rate fluctuation seems to be greater. It's better to further explain the reason for this phenomenon?

---

> ### Author Response · Authors · 2023-11-17
>
> Thank you for your insightful questions. We have prepared detailed, point-by-point responses to each query. We hope this addresses your concerns effectively.
>
> **Q1: About ablation study of frame information**.
>
> **A1:** Thanks for your suggestion. In our inital submision, we analyzed the effectivenes of coding information from previous frames (e.g., distortion $\hat{D}_{t-1}$, bitrate $\hat{R}_{t-1}$ and $\lambda_{t-1}$).
> Experimental results show that if we remove the coding information, the performance of rate allocation results for the current frame will drop (denoted as Ours w/o reference), as shown in Fig. 6. In details, the BD-rate performance will drop by 2.91%.
> Furthermore, we perform new experiments for the ablation study. Specifically, if we remove the coding information form reference frame in the rate implementation network, the trainig process will be unstable and the model will be unable to be used in rate control. Besides, if we remove the residual image in the rate implementaion network, the average rate error on DVC for HEVC Class B, C, D, and E datasets respectively rise to 8.44%, 5.31%, 12.19%, and 15.40%. And the model is closely to unuseble.
> We will provide more details in the revised version and make it more clear.
>
>
>
> **Q2: About the impact of different miniGoPs.**
>
> **A2:** Thanks for your suggestion. We made more experiments to analyze the impacts of miniGoPs in the following table. Specifically, if the miniGoP size is set to 2, BDrate performance on the HEVC Class B dataset decreases by 6.72%. If it is set to 8, the final performance increases by 0.12%. However the model parameter size significantly increases, and the training time is almost doubled. So, considering the tradeoff between the number of parameters, training time, and compression performance, the miniGoP size of our method is set to 4. We will add this analysis in our revised paper.
>
> | Size of miniGoP |  2   |  4   |   8   |
> | :-------------: | :--: | :--: | :---: |
> |   BD-Rate (%)   | 6.72 |  0   | -0.12 |
> | Parameter size  |  0.11M   |  0.44M  |   1.77M   |
>
>
>
>
>
> **Q3: About the quality fluctuation and period.**
>
> **A3:** Thanks for your question. Yes, the quality fluctuation period is related to the size of miniGoP.  The reason is that we assign different weights for each frame in a miniGoP, as a result, we have some quality fluctuation. Additionally, since the weights in different miniGoPs share similarities due to consistent video information, we can observe periodic fluctuations in bitrates.
>
>
>
> **Q4: About the ablation study of rate allocation in Fig. 7.**
>
> **A4:** Thanks for your suggestion. In our method, the rate allocation network tries to allocate different bitrates for each frame and achieve better rate-distortion performance. Therefore, it will assign more bits to important frames inside each miniGoP. In contrast, the method without rate allocation in our experiment will assign the same bitrate for each frame. Therefore. it is expected to observe a larger rate fluctuation for the proposed method. Besides, it should be highlighted that our rate allocation approach will significantly improve the compression performance (9.35% BD-rate gain on HEVC Class B dataset) compared with allocating same bitrate for each frame.

---

### Official Review · Reviewer_Tpyi · 2023-10-31

**Soundness:** 2 fair
**Presentation:** 4 excellent
**Contribution:** 3 good
**Rating:** 8
**Confidence:** 4

**Summary:**

The paper presents a new method for rate control for neural video compression. The method works by adding two new modules to standard learned video compression architectures. The first module is a "rate allocation" module, which attempts to get the average rate for a mini group of pictures to match the overall target rate specified by the user. The second module is a "rate implementation" module, which outputs frame-dependent lambda parameters for controlling the trade-off between rate and distortion. In numerical experiments the paper shows that the new rate control module effectively alters the rate for a suite of learned video compression methods from previous papers. Furthermore, the rate control scheme actually yields an improvement in BD-rate performance for all the methods.

**Strengths:**

1. The paper introduces a new method for rate control, which is a notable open problem in the field of learned compression.
2. The proposed rate control method allows some adaptability between frames so the overall codec can hit the target rate.
3. The proposed rate control method outperforms previous hand-crafted rate control methods applied to learned video compression. About a 10% compression gain is observed for most models.
4. The proposed rate control method can be applied to existing neural codecs. The paper demonstrates its application to four relevant methods from the last few years.
5. The paper is clearly presented and is easy to follow.

**Weaknesses:**

My main concern is the paper does not seem to consider all relevant literature, particularly the ELF-VC method for rate control with one-hot coded label maps (Rippel, 2021). ELF-VC is a number of years old at this point and fairly well cited, but it is not referenced in the present paper. The Rippel method would use integer-based quality levels, which is essentially identical to the standard in traditional video codecs. The present method allows specific rate targeting, which is more advanced, but still I think previous methods for rate control should be considered.

Rippel, Oren, et al. "Elf-vc: Efficient learned flexible-rate video coding." Proceedings of the IEEE/CVF International Conference on Computer Vision. 2021.

**Questions:**

1. Did you consider simple one-hot label maps as an alternative rate control mechanism? Even classical codecs are typically controlled by "quality level" parameters rather than target rates, so the rate targeting mechanism in the present work is non-standard.
2. Why does the hyperbolic model accuracy improve as the frame index increases?
3. Does the rate control method work out-of-domain?

---

> ### Author Response · Authors · 2023-11-17
>
> Thank you for your insightful questions. We have prepared detailed, point-by-point responses to each query. We hope this addresses your concerns effectively.
>
>
>
> **Q1: About previous ELF-VC work(Rippel, 2021)**
>
> **A1:** Thanks for bringing this paper to us. We will cite and discuss this paper in the revised version. It should be mentioned that ELF-VC proposed a video compression model with a variable rate coding scheme. However, it cannot realize a precise rate control. For example, if we want to compress a video 1Mbit/s, our approach can produce the corresponding $\lambda$ for each frame with optimal rate-distortion performance. However, ELF-VC work needs to search the reasonable quality-level for each frame through multi-pass coding, which is time-consuming. Therefore, our work is different from ELF-VC work.
>
>
>
> **Q2: About the one-hot label map mechanism.**
>
> **A2:** Thanks for your suggestion. We want to argue that the proposed rate control solution is NOT non-standard and a lot of traditional codecs have supported this important feature. For example, the practical codecs x264[1] and x265[2], support both VBR(Variable Bit Rate) mode and CBR(Constant Bit Rate) mode, which offer accurate control over the bit rate as our approach. For the reference software for H.265[3], they also provide the same rate control function as ours. This important feature can ensure that the codec achieves optimal performance for the given bandwidth.
>
> We believe that using the simple one-hot label maps approach can achieve compression results at different bitrates and qualities. However, in scenarios where there is a limited transmission bandwidth constraint, this approach may not accurately encode the video to meet a specific bitrate and satisfy the bandwidth requirement. Therefore, we investigate the neural rate control in this paper for the learned video codec.
>
> [1] http://ftp.videolan.org/pub/videolan/x264/
>
> [2] http://ftp.videolan.org/pub/videolan/x265/
>
> [3] Sullivan, Gary J., et al. "Overview of the high efficiency video coding (HEVC) standard." *IEEE Transactions on circuits and systems for video technology* 22.12 (2012): 1649-1668.
>
>
>
> **Q3: About the accuracy improvement of hyperbolic model.**
>
> **A3:** Thanks for your question. Different videos exhibit distinct content characteristics, and therefore, their corresponding hyperbolic model parameters also vary. In the traditional hyperbolic model mapping from R to \lamdba, a predefined set of parameters is usually used. During the encoding process, these model parameters are iteratively adjusted based on prediction errors. Consequently, during the initial stages of encoding, the hyperbolic model parameters may not necessarily adapt well to the current video content, resulting in relatively larger prediction errors during the early encoding process. As adjustments to the \alpha and \beta parameters in the hyperbolic continue throughout the encoding process, the prediction error gradually decreases. And we will add this explanation to the paper.
>
>
>
> **Q4: About the model's working domain.** "Does the rate control method work out-of-domain?"
>
> **A4:** Thanks for your question. But we feel sorry that we are a little bit confused about your question about "domain".
>
> If you are referring to issues with the training and testing dataset, the answer is that rate control method works out-of-domain. Our training dataset is vimeo-90k and BVI-DVC dataset. In vimeo-90k dataset, each set of data contains a continuous sequence of seven frames, with a resolution of 448x256. BVI-DVC dataset consists of 800 video sequences with varying resolutions, each sequence containing 64 frames. The test datasets include HEVC (Class B, C, D, E), UVG, MCL_JCV datasets. HEVC dataset conclude videos with different resolution and frame rates. Its resolution ranges from 416x240p to 1920x1080p. The resolution for UVG dataset and MCL_JCV dataset is 1920×1080. As you can see, our training data and testing data have very different properties and the test data covers various video domains.
>
> If we do not correctly understand your point, please let us know. We will respond to you as soon as possible.

---

> > ### Comment · Reviewer_Tpyi · 2023-11-20
> >
> > I would like to thank the authors for addressing my comments on the paper. I am satisfied with the answers and have changed my recommendation to accept the paper.

---

> > > ### Author Response · Authors · 2023-11-23
> > > **Response to Reviewer Tpyi**
> > >
> > > Thank you for your positive feedback and recommendation of our paper for acceptance. We sincerely appreciate your valuable comments, and we will incorporate your suggestions to enhance our work.

---

### Official Review · Reviewer_c8Ca · 2023-11-03

**Soundness:** 3 good
**Presentation:** 3 good
**Contribution:** 2 fair
**Rating:** 5
**Confidence:** 5

**Summary:**

This paper proposes a learnt architecture for rate control in video compression. This is achieved by the rate control module to automatically assign the weights for consecutive frames and then allocate bit-rates according to the budget. Then, a bit-rate implementation network is proposed to output the hyper-parameter \lambda to achieve the RD trade-off, in which the allocated bit-rate can be truly consumed. Since the bit-rate allocation and implementation modules are learnt by two stages, the proposed method is the plug-and-play method to control the bit-rates for different learnt video compression codecs. The experimental results have verified the effectiveness of the proposed method.

**Strengths:**

1. The learnt rate control module has been proposed in this paper, which is able to control the bit-rate in a plug-and-play style.
2. The bit-rate implementation network also contributes to the rate control of learnt video compression method.
3. The experimental results exhibit the effectiveness of the proposed plug-and-play method, against 4 learnt video compression methods.

**Weaknesses:**

1. This paper claims that the proposed method is the first fully neural network for rate control in learnt video compression. Please elaborate more on this, given that many learnt methods available to achieve the rate control for learnt video compression, e.g., [1].
2. The proposed method is trained in separate stages, which are with limited contributions by my side. It is the fact that many rate control methods aim to fit closed-form mathematical models, e.g., R-\lambda, R-\rho and R-Q models. The proposed bit-rate allocation module essentially can be regarded to learn to implicitly fit the R-\lambda model. If so, the comparison with closed-form models should also be reported, for example, against HEVC and VTM as also mentioned in the paper.
3. I am surprised by the reported experimental results, whereby the RD performances could be further improved by adding rate control scheme. The target bit-rates were obtained by optimizing R+\lambda D with constant \lambda, which means the achieved D now should be the lowest distortion given the target bit-rate R and constant \lambda. The proposed method controls the bit-rates by adjusting \lambda, which in my opinion is supposed to perform inferior to the non-rate-control method. Why adding rate control can further improve the RD performance?

[1] Mao, Hongzi, et al. "Neural Rate Control for Video Encoding using Imitation Learning." arXiv preprint arXiv:2012.05339 (2020).

**Questions:**

Please see my weakness.

---

> ### Author Response · Authors · 2023-11-17
>
> Thank you for your insightful questions. We have prepared detailed, point-by-point responses to each query. We hope this addresses your concerns effectively.
>
> **Q1: About the "first" claim of our work**.
>
> **A1:** Thank you for your question. Mao et al.'s work attempts to address the rate control problem in **traditional video compression** methods like VP9 and H.264 by utilizing neural networks. In contrast, our work is the first fully neural network-based approach for rate control in **learned video compression** such as DVC and FVC.
>
> Furthermore, from a technical standpoint, Mao et al.'s work employs a neural network to learn the QP for each frame, which determines the rate allocation. The optimal QP target is obtained through an evolution strategy algorithm. On the other hand, we use a neural network for both rate allocation and implementation stages, optimizing the entire framework directly using Rate-Distortion loss without relying on time-consuming labeled optimal QP datasets as done in Mao et al's work.
>
> In summary, our study presents distinct differences from related research, reinforcing our unique contributions. These distinctions will be further elaborated upon in the revised version of our paper.
>
>
> **Q2: About the separate training stages.**
>
> **A2:** (1) In our approach, we first train an accurate rate implementation network. Then, for the training of the rate allocation network, the whole framework, including the rate implementation network, is involved in the optimization and we directly use the rate-distortion loss in this stage (Eq.(5) in the initial paper).
>
> Therefore, although we progressively train the rate implementation network and the rate allocation network, the optimization of the rate allocation also relies on the rate implementation and the learned video codec, which ensures an accurate rate control policy.
>
> (2) The rate implementation is indeed optimized to accurately map the target rate to the encoding parameter $\lambda$. Our approach is superior to traditional methods, as demonstrated in our initial version on Page 8. Specifically, we conducted a comparison by implementing one traditional mathematical method, the hyperbolic R-$\lambda$ model on the DVC codec. When testing on HEVC C and D sequences, the traditional method resulted in bitrate errors of 7.72% and 8.43%, respectively. In contrast, our method achieved significantly lower bitrate errors of only 1.18% and 1.91%. These results are also shown in Fig.5, confirming that our approach has better rate implementation capability due to its incorporation of spatial-temporal information using neural networks.
>
>
>
> **Q3: About the RD performance gain in the experiments.**
>
>
> **A3:** Thank you for your question. Most existing video compression methods use the same $\lambda$ value for each frame and lead to a cumulative error problem. In contrast, our proposed rate control method learns to assign different bitrates to each frame by utilizing different $\lambda$ values. This not only achieves effective rate control but also reduces the issue of error propagation. As a result, our approach can improve compression performance.
>
> We agree with your opinion on rate control in **traditional video coding**. In this context, implementing rate control can actually decrease overall compression performance. This is because traditional video codecs typically utilize a hierarchical quality structure and do not experience error propagation issues. Therefore, introducing rate control in this scenario will not result in performance improvements.
>
> Moreover, if the existing learned video codec already possesses an appropriate hierarchical quality ($\lambda$) structure, our method will not provide any additional gains in compression performance. However, it is important to emphasize that the primary contribution of our paper lies in offering an accurate rate control solution that is independent of the learned codec baselines, regardless of whether they have a hierarchical quality structure or not.

---

### Public Comment · ~Tongda_Xu1 · 2023-11-25
**Would you like to add discussion on a concurrent work?**

Hi, authors of *Neural Rate Control for Learned Video Compression*,

Thanks for writing this great work, I like it a lot and I wish I could have done it. I do rate control and I know how hard it is to make it work. I would like to mention that there is a concurrent work named *Bit allocation using optimization* that is just published in July, ICML 2023. This work also deal with rate allocation for neural video compression. It is __very very relevant__ to your submission. I do not think it is necessary to compare to it, but I think adding a discussion on it makes your submission more complete.

Xu, T., Gao, H., Gao, C., Wang, Y., He, D., Pi, J., ... & Zhang, Y. Q. (2023, July). Bit allocation using optimization. In International Conference on Machine Learning (pp. 38377-38399). PMLR.

Thank you for reading this comment and I hopr this great work can be accept!

---

### Meta-Review · Area_Chair_1zqA · 2023-12-10

**Metareview:**

The paper received somewhat mixed but overall positive ratings (8, 8, 6, 5). The reviewers acknowledged that the paper tackles an important open problem in video compression and that the proposed approach is sufficiently novel with strong empirical performance. Most of the initial concerns are adequately addressed by the authors in their rebuttal. Given the merits of the paper, we recommend acceptance.

**Justification For Why Not Higher Score:**

There were several comments regarding missing references, indicating that the paper could benefit from a more thorough review of existing literature. Additionally, while neural video compression is certainly within the scope, it might not reach the wider audience of ICLR.

**Justification For Why Not Lower Score:**

The paper tackles an important problem and proposes a convincing approach supported by good performance.

---

### Decision · Program_Chairs · 2024-01-16

Accept (poster)